# Depth Quality Improvement with a 607 MHz Time-Compressive Computational Pseudo-dToF CMOS Image Sensor [note 1]

**DOI:** 10.3390/s23239332

**Published:** 2023-11-22

**Authors:** Anh Ngoc Pham, Thoriq Ibrahim, Keita Yasutomi, Shoji Kawahito, Hajime Nagahara, Keiichiro Kagawa

**Affiliations:** 1Graduate School of Integrated Science and Technology, Shizuoka University, Hamamatsu 432-8011, Japan; pham.ngoc.anh.17@shizuoka.ac.jp (A.N.P.);; 2Research Institute of Electronics, Shizuoka University, Hamamatsu 432-8011, Japan; 3Institute of Datability Science, Osaka University, Suita 565-0871, Japan

**Keywords:** pseudo-dToF imaging, charge-domain time-compressive sensing, multipath interference

## Abstract

In this paper, we present a prototype pseudo-direct time-of-flight (ToF) CMOS image sensor, achieving high distance accuracy, precision, and robustness to multipath interference. An indirect ToF (iToF)-based image sensor, which enables high spatial resolution, is used to acquire temporal compressed signals in the charge domain. Whole received light waveforms, like those acquired with conventional direct ToF (dToF) image sensors, can be obtained after image reconstruction based on compressive sensing. Therefore, this method has the advantages of both dToF and iToF depth image sensors, such as high resolution, high accuracy, immunity to multipath interference, and the absence of motion artifacts. Additionally, two approaches to refine the depth resolution are explained: (1) the introduction of a sub-time window; and (2) oversampling in image reconstruction and quadratic fitting in the depth calculation. Experimental results show the separation of two reflections 40 cm apart under multipath interference conditions and a significant improvement in distance precision down to around 1 cm. Point cloud map videos demonstrate the improvements in depth resolution and accuracy. These results suggest that the proposed method could be a promising approach for virtually implementing dToF imaging suitable for challenging environments with multipath interference.

## 1. Introduction

Time-of-flight (ToF) depth imaging calculates the distance of an object through illuminating a scene with a modulated light source and measuring the travel time of light emitted from the camera system to the object and reflected back to the camera. This technology enables high-accuracy and high-frame-rate 3D imaging using a compact system consisting of a CMOS image sensor together with an active modulated light source. Because of these advantages, ToF measurement has been realized in a wide range of machine vision applications, such as gesture capture and behavior recognition, and visual positioning and guidance [1,2,3]. However, the current mainstream ToF sensors can suffer from a relatively low spatial resolution [4,5] due to their large digital circuit areas, or scene-dependent errors due to multipath interference (MPI) problems [6]. Light from multiple paths, caused by interreflections and scattering, enters the same pixel, resulting in depth errors. Therefore, ToF sensors with both high spatial resolution and robustness to MPI are desired for realizing simpler and more accurate depth camera systems.

There are two major methodologies for traditional ToF depth imaging: direct ToF (dToF) and indirect ToF (iToF). The dToF sensors directly measure the ToF of incident light and express it as a waveform of reflected light intensity. This approach necessitates fast-response photodetectors with single-photon sensitivity, and in most cases, single-photon avalanche diodes (SPADs) [7,8] are utilized. The dToF sensors are relatively robust against MPI because it is easier and simpler to compensate for MPI since the entire reflected light waveform is observed. However, they yield lower spatial resolution because of the large area required for complex circuitry such as time-to-digital converters (TDCs) and histogram builders. In contrast, iToF sensors [9] use the correlation between the incident light and the time windows to estimate the ToF or the temporal phase difference. The correlation is given through the number of modulated photoelectrons corresponding to each time window. The iToF sensors are suitable for large pixel counts thanks to their simple circuits. However, they are susceptible to MPI due to the loss of reflected light waveform information after correlation. In addition, the multiple measurements for different time window timings often used in the amplitude modulation continuous wave (AMCW) method lead to motion artifacts in dynamic scenes [10]. Therefore, with the conventional approaches, it is difficult to achieve an appropriate balance among high spatial resolution, high accuracy, and tolerance to MPI.

In a paper at IISW 2023 [11], we presented a novel method called pseudo-direct time-of-flight (pseudo-dToF) depth imaging, which provides the advantages of both dToF and iToF image sensors. This method utilizes high-speed charge modulators for iToF, allowing for high-spatial-resolution imaging like conventional iToF image sensors [12,13]. Random exposure codes are applied to time windows in demodulation to perform time-compressive sensing in the charge domain [14,15,16]. This key technique allows us to acquire the entire light waveform in a single shot like those obtained using dToF sensors. Therefore, pseudo-dToF imaging provides high accuracy, robustness to MPI, and motion-artifact-free depth measurement. However, the depth precision is limited by the maximum modulation frequency of the time window, which corresponds to the bin size in the dToF. To overcome this drawback without any hardware modification, we presented two approaches. First, a sub-time window is introduced to halve the time resolution. Second, the sensor responses are oversampled to increase the number of reconstructed data points, and quadratic fitting is performed in the depth calculation to refine the distance results. The fitting is similar to the technique used in dToF to realize sub-bin depth resolution [17,18].

In this paper, we additionally describe the detailed sensor configuration and the reconstruction signal processing. ToF measurement videos are also presented, visually confirming the benefits of the proposed method in enhancing the time resolution without changing the hardware. Appendix A are available as Appendix A.

## 2. Pseudo-dToF Depth Imaging

In our scheme, ToF imaging can be divided into three phases: pre-measurement, sensing, and signal reconstruction.

In the pre-measurement phase, all data used for the subsequent signal reconstruction are prepared. Exposure codes are generated, and the instrument response functions (IRFs) of the imaging optics, the light source, and the image sensor are measured in advance. Note that this preparation step needs to be performed only once for all measurements. During the sensing phase, the camera captures temporal signals of the incident light emitted from a synchronized laser and reflected to the camera through a single-aperture lens. The image sensor consists of macro-pixels based on charge modulators, as shown in Figure 1. During the image acquisition, exposure codes, i.e., temporal electronic shutters, are provided for each tap of the charge modulator in each pixel. Thus, incident light signals are temporally compressed in the charge domain, yielding four temporally compressed images in one shot. In the reconstruction phase, the input optical waveforms for all sub-pixels are reconstructed. Through solving an inverse problem based on sparsity regularization, temporally sequential images or transient images of light are obtained. The depth calculation is then processed as in normal dToF imaging. The arrival times of the peaks of the light waveform are detected with quadratic curve fitting, and finally, they are converted to a depth image using the speed of light.

### 2.1. Multi-Tap Macro-Pixel CMOS Image Sensor

This sensor utilizes a charge modulator-based iToF image sensor, which enables a small pixel size or high-spatial-resolution imaging. Figure 2 shows the sensor block diagram and the pixel structure. The image sensor is composed of a pixel array, vertical and horizontal scanners, on-chip ADCs, and a readout circuit as a typical CMOS image sensor. The peculiar circuits in this sensor include the lab-designed macro-pixel array, the shutter controller, and the charge modulator drivers that control the signal accumulation in the macro-pixels. Every macro-pixel consists of four sub-pixels, each of which is a four-tap lateral electric field charge modulator (LEFM) [19]. The LEFM is composed of a photodiode, and four charge transfer gates G[i] and four storage diodes FDi (i=1,…, 4), configured as a pair referred to as a tap. Here, the index i identifies the tap. In the capture phase, the shutter controller outputs the pre-loaded exposure codes, which are applied to the transfer gates of each pixel via the driver circuit operating at a high-speed clock. All sub-pixels perform the charge modulation according to the exposure codes. When the exposure time is finished, the captured images are read out by the ADCs and the readout circuit.

Transfer of the generated photoelectrons depends on the exposure codes assigned to each transfer gate. These exposure codes, representing the time window function, are binary, as shown in Figure 3. A value of “1” indicates the transfer of photoelectrons, while “0” signifies no charge to be transferred. Through employing temporally randomized exposure codes, the incident light signals are temporally compressed in the charge domain. Considering that a time-variant optical signal $x\left(t\right)$ is subjected to a designated time window function $w_{i}\left(t\right)$ at tap i, the output integrated charge value $y_{i}$ can be defined through correlation:(1)$$y_{i}=\int_{0}^{T_{exp}}w_{i}\left(t\right)\;\cdot \;x\left(t\right)dt$$
where $T_{exp}$ is the exposure time for each frame.

Table 1 shows the specifications of the prototype image sensor. In our previous research, we confirmed that the sensor was driven at a maximum modulation frequency of up to 303 MHz [19]. This high-speed operation of the LEFM pixels allows for high temporal resolution in ToF imaging. Also, no signal processing circuits are necessary for compressive sensing because compression processing is done through charge transfer. The only additional circuit is the shutter controller that generates exposure code, which can be implemented as a relatively small digital circuit, thus enabling high spatial resolution.

### 2.2. Compressive Sensing

Compressive sensing [20,21] is an efficient sampling method that reconstructs more data points from fewer samples when the original signal is sparse. We employ this principle to reconstruct the entire incident light waveform from the compressed output images. This allows pseudo-dToF imaging to obtain dToF-like signals that are robust against MPI, enabling high depth accuracy while being comparable with the conventional iToF image sensors in terms of their architecture and components. Note that compressed images are obtained in a single shot, which makes our sensor free of motion artifacts.

Time-compressive sensing and signal reconstruction are mathematically described as follows. Considering the characteristics of the imaging optics and charge modulations, Equation (1) can be expanded to:(2)$$y_{i}=\int_{0}^{T_{exp}}\left\{w_{i}\left(t\right)\;*\;h\left(t\right)\right\}\;\cdot \;\left\{p(r)\;*\;x\left(r,t\right)\right\}dt$$

Here, $w_{i}\left(t\right)$ represents the exposure code, $h\left(t\right)$ is the sensor’s temporal IRF or sensor impulse response, p(r) is the spatial IRF of the imaging optics, $x\left(r,t\right)$ shows incident light intensity at $\left(r,t\right)$ where r indicates a two-dimensional spatial coordinate. The ∗ operator indicates convolution. The exposure codes are repeatedly applied during the exposure duration to obtain high photosensitivity through multiple exposures. Collectively, the exposure code, the sensor’s IRF, and the spatial IRF can be referred to as the spatio-temporal observation matrix A. Equation (2) is then expressed in discrete values as follows:(3)y=Ax

In cases where the detected signal y is lower in dimensionality than the original input signal x, the temporal information of x is compressed. Retrieving the input signal x from y becomes an ill-posed problem. However, if x is *K*-sparse, implying that only *K* elements have non-zero values, estimating the solution for x becomes feasible through optimizing the L1 norm. In this research, we estimate the closest solution for x through minimizing the total variation, as shown in:(4)$${\hat{x}}^{\left(TV\right)}=\arg\underset{x}{\min}\sum_{i}{\left\|D_{i}x\right\|}_{1},\;s.t.\;y=Ax$$

Here, $D_{i}$ represents a spatio-temporal differential operator. This process is performed by TVAL3 [22], a compressed sensing solver, using an iterative method. Matrix A is selected as a random matrix to minimize the correlation between the row and column vectors of the observation matrix [20]. As matrix A significantly influences the quality of the reconstruction, a random exposure pattern is generated and optimized in the pre-measurement phase. In this study, three different images were used for the reconstruction in order to mitigate the influence of the scene on the reconstruction results. For each image, we generated the exposure code at random, reconstructed the image, and iterated this process 30 times. The exposure code yielding the image with the highest average PSNR was selected. Note that due to the structure of the multi-tap macro-pixel, creating an ideal random exposure code with 16 taps may not be feasible. Consequently, it becomes necessary to create a hypothetical random pattern under specific generating conditions. For example, in a sub-pixel, only one tap can be turned on at any given time, and there must always be one active tap. Based on this principle of compressive sensing, the reflected light waveforms x of all sub-pixels are reconstructed from the four compressed images y and the pre-measured observation matrix A.

Depth information is calculated after the reconstruction of transient images. To generate a depth map, the reconstructed waveform of each pixel is analyzed to identify the peak arrival time, which corresponds to the round-trip ToF. Finally, it is converted into a depth image using the speed of light. In pseudo-dToF imaging, the measurement range and depth resolution are dependent on the bit length of the exposure code and the frame rate of the reconstructed transient images. In the simplest situation, the temporal resolution is determined using the minimum time window width of the exposure code, which corresponds to the bin width of the dToF histogram. The measurement range is calculated through multiplying the depth resolution by the bit length. For example, if our sensor operates at 303 MHz and uses a 32-bit exposure code, the temporal resolution is 3.3 ns (or 0.5 m in depth), and the measurable distance is 16 m. Note that sub-time-window interpolation is often used to improve the depth resolution.

## 3. Improvement of Depth Resolution

So far, we have successfully performed ToF imaging at an operating clock frequency of 303 MHz [19]. This allowed a minimum time window with a duration of 3.3 ns, which corresponds to a 0.5 m resolution when only the frame position that gives the highest pixel value is considered. Generally, higher depth resolution can be achieved simply through increasing the operating frequency of the charge modulator. However, it was challenging to achieve a minimum time window duration below 3.3 ns due to the performance limitations of the charge modulators. Driving the charge modulation at 303 MHz is quite fast compared to the fastest modulation speed of 320 MHz [23], and realizing a higher frequency requires not only advanced fabrication technology but also an improved modulator and driver structures. To overcome these restrictions, the following two approaches were employed to obtain higher depth resolution without any hardware modifications.

### 3.1. Sub-Time Window

Here, a sub-time window of the exposure code is introduced to enhance the temporal resolution without affecting the minimum time window duration, which is determined by the modulator’s performance. To implement the sub-time window, we double the frequency of the on-chip phase locked loop (PLL) from 303 MHz to 607 MHz while maintaining the minimum time window duration at 3.3 ns (equivalent to two clocks). As shown in Figure 4, the rising and falling edges of the time window can be shifted by one clock unit. This shift is half of the minimal time window, resulting in halving of the temporal resolution to 1.65 ns.

### 3.2. Oversampling and Peak Detection with Fitting

Another approach to improve the temporal resolution is to oversample the IRF. When reconstructing the transient images, the quantity of reproduced frames is equal to the number of data points in the IRF $h\left(t\right)$ or the observation matrix A. Consequently, increasing the number of data points in the pre-measured observation matrix will subsequently increase the number of reconstructed frames, resulting in finer temporal resolution.

Normally, the IRF is sampled once per bit of the exposure code. However, in this study, we sub-sampled 10 points per bit (10× sampling), as depicted in Figure 5. A light waveform is then reconstructed with improved temporal (or depth) resolution. Since the signal waveforms can be obtained as in dToF imaging, we perform fitting to refine the depth. After the peak position in the waveform is detected, the five data points around the peak are considered in order to fit a quadratic curve to estimate the exact signal peak. Note that a similar depth calculation technique based on fitting or filtering is utilized in dToF to realize sub-bin depth resolution [17,18].

The depth accuracy and precision of the pseudo-dToF method are significantly improved because higher-density light waveforms through oversampling are utilized in depth estimation with fitting. This approach can distinguish and detect each peak separately even when there are multiple peaks due to MPI. It is expected that the pseudo-dToF method will become applicable even in challenging environments with MPI. Note that the depth resolution is improved without altering the image sensor hardware or the total measurement time, although the reconstruction process takes longer.

## 4. Experimental Results

For all experiments in this study, 32-bit long exposure codes corresponding to a range of 16 m at a clock frequency of 606 MHz were selected to accommodate the objects in the experimental scenes. The exposure code length determines the depth range and the compression ratio. As a result, a moderate compression ratio of 8× in the total spatio-temporal sampling points (2× in the temporal domain and 4× in the number of pixels) was achieved. In terms of the compression ratio, image reconstruction becomes more challenging at higher compression ratios, meaning longer exposure codes.

Firstly, we verified the feasibility of driving the sensor with doubled speed at 607 MHz while maintaining the minimum time window width. Figure 6 shows the temporal system response measured when this sensor was operated at 607 MHz. The exposure code was a 32-bit random binary code, and a short-pulse semiconductor laser (PicoQuant, Belin, Germany, LDH-IB-450-M-P, wavelength of 443 nm) with a pulse width of 228 ps was used. The pixel values were averaged for each tap, and dark frames were subtracted to reduce the dark noise level. It can be observed that charge modulation was successfully performed.

To evaluate the improvement of depth resolution with the sub-time window approach, we compared the signal peak separation performance under MPI conditions. In this experiment, interference light was introduced through placing a weakly diffusive plastic sheet in front of a panel serving as an object, with the black letters “SU” on it, as shown in Figure 7a. The letters look blurry due to the diffuser. The distance between the two objects was set to 0.65 m and 0.40 m, and experiments were conducted with the sensor operating at 303 MHz and 607 MHz. An ordinary single-aperture imaging lens (Edmund Optics, C series VIS-NIR fixed focal length lens, focal length of 16 mm, f/1.4) was used. The light source and exposure codes were the same as in the measurements shown in Figure 6. The actual image readout frame rates were 16.28 fps at 303 MHz and 18.34 fps at 607 MHz.

Next, a ToF imaging experiment was conducted to verify the effectiveness of oversampling for a ratio of 10. Note that the time/depth resolutions at 303 MHz and 607 MHz (with normal sampling), and 607 MHz (with 10× sampling) are 3.3 ns/0.5 m, 1.75 ns/0.25 m, and 0.175 ns/0.025 m, respectively. A pulsed semiconductor laser (Tama Electric Inc., Hamamatsu, Japan, LDB-300C) with a wavelength of 660 nm and an FWHM of 2.55 ns was used. One hundred images were averaged to improve the SNR. The actual image readout frame rates were 8.57 fps at 303 MHz and 12.13 fps at 607 MHz. The image processing was performed using MATLAB R2020b on an Intel^®^ Core™ i7-9700 CPU (Intel, Santa Clara, CA, USA) running at 3.00 GHz equipped with 32 GB of memory. The image reconstruction time is approximately *N* times longer with *N*× sampling compared to without oversampling. This increase in time is primarily because the column dimension of the observation matrix ***A*** in Equation (3) becomes *N* times larger. Specifically, the reconstruction time for an image captured with 32-bit exposure codes is approximately 200 s without oversampling, and averages about 1800 s with 10× sampling. Notably, no acceleration through parallel processing was utilized in the reconstruction. Figure 8 shows the configuration of the targets.

Figure 9a,b compares the depth maps for each condition in frontal and right-side views. Figure 9c shows the reconstructed optical waveforms and their fitting curves with a quadratic approximation. The mean and standard deviation of the depths are quantitatively compared in Table 2 to demonstrate the effectiveness of the proposed approaches. In the result at 607 MHz, 10× sampling has the closest mean depths to the actual values and the smallest standard deviation of about 1 cm.

ToF videos were captured using the proposed method. Two setups are shown in Figure 10a and Figure 11a, one without and the other with MPI. In the first acquisition, a man was observed walking toward the camera system in the background with obstacles. These shots were filmed in a dark room, using the same equipment as the experiment shown in Figure 8. The actual image readout frame rate was 7.68 fps. The videos in Figure 10b,c display the reconstructed 3D point cloud map without image averaging, at a PLL clock frequency of 303 MHz for normal sampling and 10× sampling. When we compare the two videos, oversampling and quadratic fitting yield cleaner, less noisy, and more defined outcomes. The object shapes in Figure 10c are more distinct and precise. These enhancements can significantly benefit subsequent depth-related tasks, like segmentation or gesture recognition, for improved quality. Furthermore, the superiority of pseudo-dToF imaging for dynamic scenes was confirmed since no motion artifacts were observed even for moving objects. Note that in this experiment, the man was wearing protective glasses, and the hair had low reflectivity. Hence, the head is difficult to capture at a long distance and only becomes clear when the man moves closer to the camera.

The setup for the video containing the MPI is the same as the experiment depicted in Figure 7. In this acquisition, the diffuser was positioned in front of the panel and moved vertically. The actual image readout frame rate was 17.24 fps. Both results, one with and one without oversampling, accurately captured the depth of the diffuser and the panel, even while the diffuser was moving. Figure 11c represents the result after applying oversampling and fitting, showing improved results with smoother and sharper objects. However, there is some crosstalk between the two objects, which is probably influenced by the difference between the measured and actual IRFs.

## 5. Conclusions

In this paper, we demonstrated the concept, benefits, and implementation of pseudo-dToF imaging. The iToF-based high-speed charge modulators are utilized to achieve high spatial resolution. The time-compressive sensing in the charge domain is performed through applying 32-bit random exposure codes in demodulation. Since the image reconstruction yields dToF-like output waveforms, pseudo-dToF is immune to MPI and motion artifacts. To improve the temporal resolution and estimated depth precision, two approaches are applied: a sub-time window and oversampling. While there are still some challenges, such as long processing time, vulnerability to ambient light, and so on, the proposed method remains a promising technique for applications like autonomous vehicles, and robotics in challenging environments with MPI. To further increase the compression ratio and to reduce the image reproduction time, longer exposure codes and image reproduction algorithms based on deep neural networks are being developed.

## Figures and Tables

**Figure 1 sensors-23-09332-f001:**
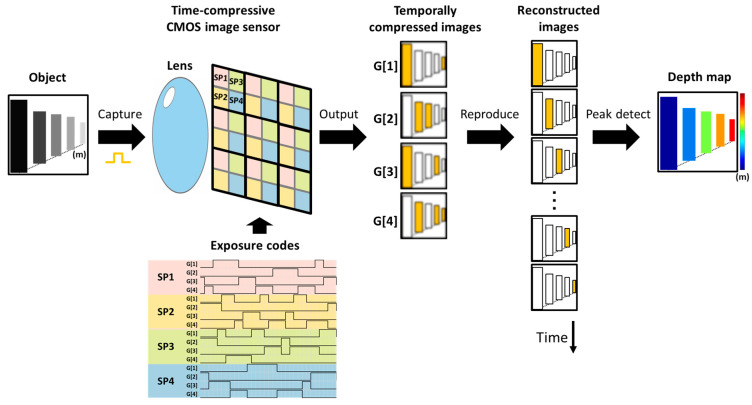
Conceptual image acquisition and reconstruction flow. Reflected light pulses from the objects enter the image sensor through the lens. The prototype image sensor is composed of a macro-pixel array, with sub-pixels SP1–SP4 constituting each macro-pixel. Temporally random exposure codes are applied to the modulators of each pixel to obtain compressed images. The light signals are then reconstructed and converted into a depth map. In this example, because each sub-pixel has four taps (G[1]–G[4]), 16 exposure codes are applied to each macro-pixel.

**Figure 2 sensors-23-09332-f002:**
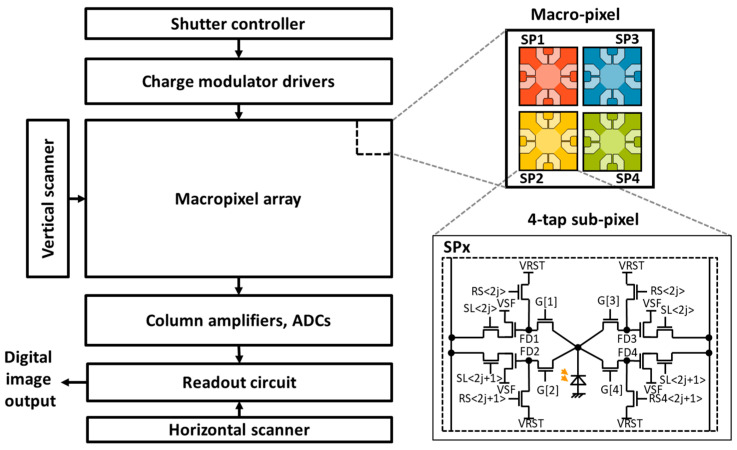
Image sensor configuration. One macro-pixel is composed of 2 × 2 four-tap sub-pixels (SP).

**Figure 3 sensors-23-09332-f003:**
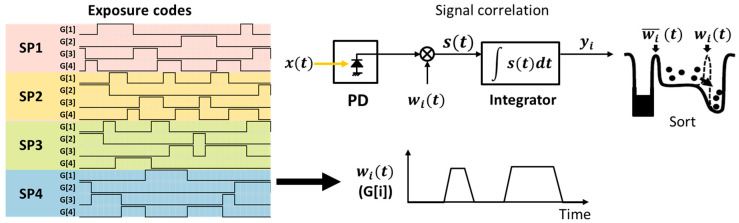
Temporal signal compression in the charge domain.

**Figure 4 sensors-23-09332-f004:**
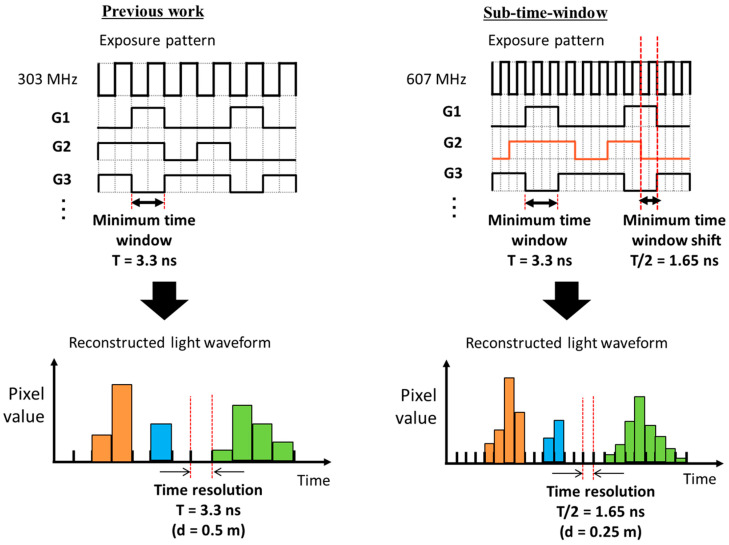
Improvement in temporal resolution through introducing sub-time window. The PLL clock frequency is doubled to 607 MHz, while the minimum time window width remains at 3.3 ns. Thus, the time window can be shifted by one clock unit, and the time resolution is halved.

**Figure 5 sensors-23-09332-f005:**
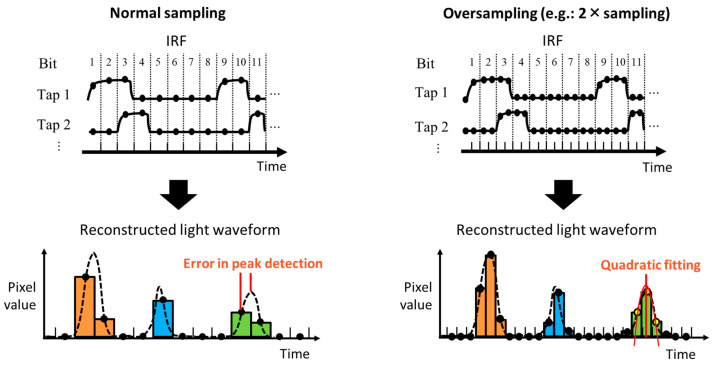
Pre-measured sensor response oversampling and quadratic fitting.

**Figure 6 sensors-23-09332-f006:**
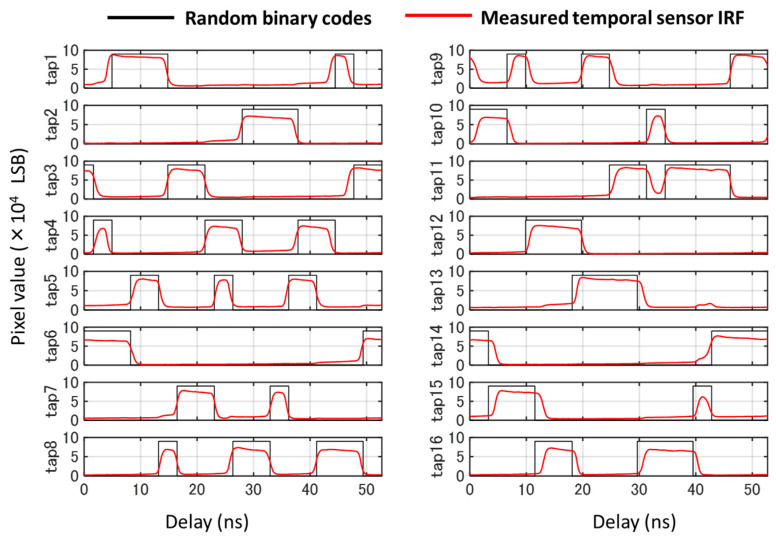
Measured response for the random exposure code at 607 MHz.

**Figure 7 sensors-23-09332-f007:**
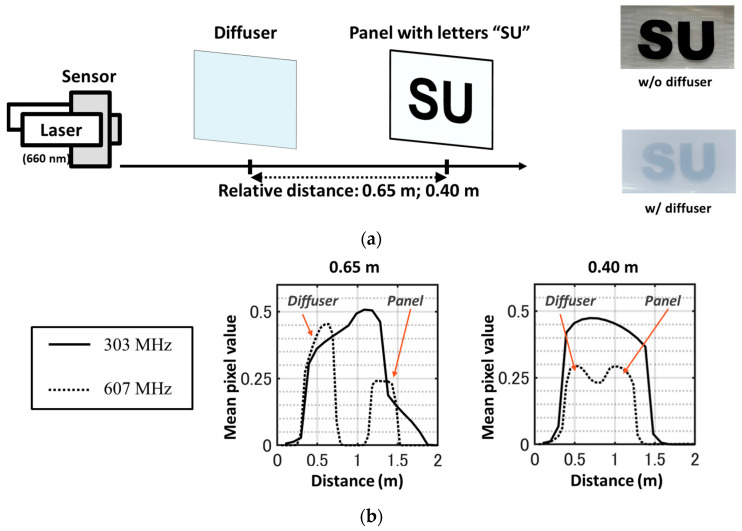
MPI interference separation at 303 MHz and 607 MHz: (**a**) experimental setup for ToF imaging with MPI; (**b**) reconstructed light waveform at relative distances of 0.65 m and 0.40 m (average of ROI 5 × 5 pixel values).

**Figure 8 sensors-23-09332-f008:**
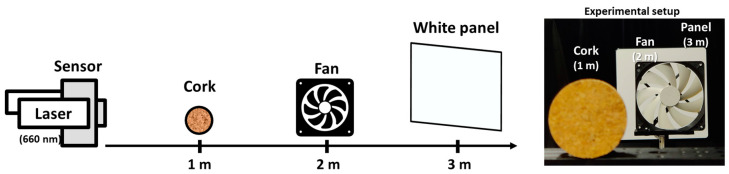
Experiment setup for single-path ToF imaging.

**Figure 9 sensors-23-09332-f009:**
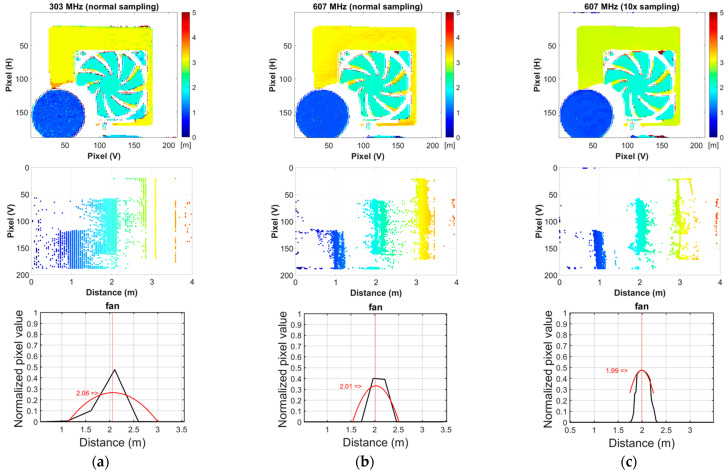
Depth maps of the front view and right-side view, and an example of reconstruction signal: (**a**) 303 MHz, normal sampling; (**b**) 607 MHz, normal sampling; (**c**) 607 MHz, 10× sampling.

**Figure 10 sensors-23-09332-f010:**
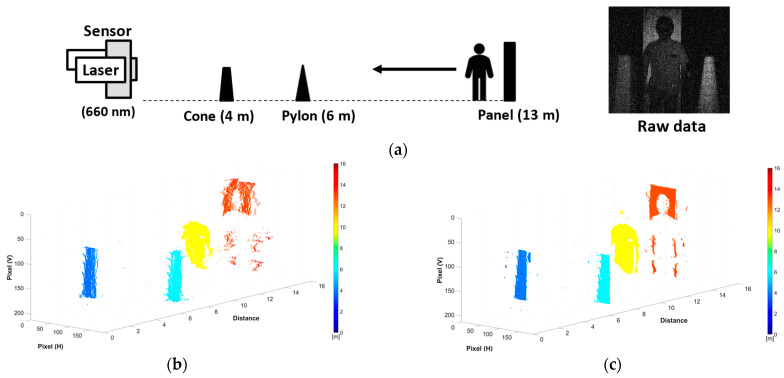
Experiment for ToF video at 303 MHz: (**a**) experimental setup and a raw captured image; (**b**) reconstructed point cloud map with normal sampling; (**c**) reconstructed point cloud map with 10× sampling and fitting. The videos of the 3D depth maps are included as Appendix A.

**Figure 11 sensors-23-09332-f011:**
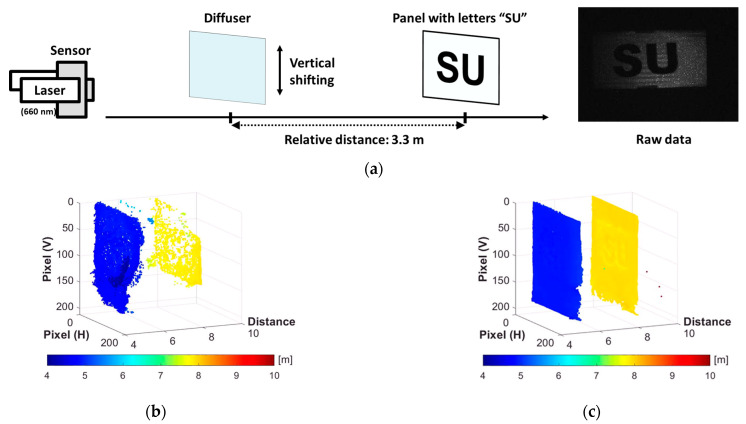
ToF video at 303 MHz in MPI condition: (**a**) experimental setup and a raw captured image; (**b**) reconstructed point cloud map with normal sampling; (**c**) reconstructed point cloud map with 10× sampling and fitting. The videos of the 3D depth maps are included as Appendix A.

**Table 1 sensors-23-09332-t001:** Sensor specifications and performance.

Technology	0.11 μm FSI CIS
Chip size	7.0 mm^H^ × 9.3 mm^V^
Macro-pixel size	22.4 μm^H^ × 22.4 μm^V^
Effective sub-pixel count	212^H^ × 188^V^
Sub-pixel count per macro-pixel	2^H^ × 2^V^
Maximum exposure code length	256 bits by 8 bits
Maximum modulation frequency	303 MHz
Maximum clock frequency	607 MHz
Maximum image readout frame rate ^1^	21 fps
Power consumption	2.8 W

^1^ Readout frame rate when the exposure time is set to zero. The actual image readout frame rate is given through the reciprocal of the formula: (exposure code bit length × bit period) × number of light pulses + readout time, where the readout time is 47.6 ms (1/21 fps).

**Table 2 sensors-23-09332-t002:** Mean and standard deviation of objects’ depths (average of 3 × 3 ROI pixels of 20 measurements).

	Real Depth (m)	303 MHz, Normal Sampling		607 MHz, Normal Sampling		607 MHz, 10× Sampling	
		Mean (m)	Std (cm)	Mean (m)	Std (cm)	Mean (m)	Std (cm)
Cork	1.00	1.04	2.53	1.01	1.17	1.00	0.94
Fan	2.00	1.66	5.59	1.98	1.14	1.99	0.72
Panel	3.00	3.03	3.67	3.08	5.60	3.00	0.57

## Data Availability

Data are contained within the article and Appendix A.

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
