# Peer review of "Depth Quality Improvement with a 607 MHz Time-Compressive Computational Pseudo-dToF CMOS Image Sensorâ€"

_sensors, 2023, doi:10.3390/s23239332_

Round 1
Reviewer 1 Report
Comments and Suggestions for Authors
This paper proposes a prototype pseudo-direct time-of-flight (ToF) CMOS image sensor with a 607 MHz time-compressive computation, which obtians high accuracy, precision, and robustness. Overall, this is an interesting topic, and the idea behind the study is well justified. Here are some issues need to be addressed:
1. Please describe how exposure codes are generated during the premeasurement phase.
2. Utilizing the oversampling method improves the measurement accuracy but increases the reconstruction time. It is suggested to discuss the relationship between the number of sampling points per bit and the reconstruction time.
3. In this article, it is stated that the image sensor does not have the disadvantage of motion artifacts of iTOF. It is strongly suggested to provide measurement results of moving objects in the experimental results.
4. The oversampling method is used to improve the measurement accuracy. How much measurement accuracy can be improved by oversampling methods?
5. There are many conceptual introductions in the article and authors should give a brief analysis of the iToF circuits of the measurement system.
Comments on the Quality of English LanguageNo
Reviewer 2 Report
Comments and Suggestions for Authors
Here are a few points that the authors could address;
* Figure 1: For the plot of "Object," please indicate what the horizontal axis represents. Is it the depth z?
* Page 3, Line 109: highe-spatial-resolution --> high-spatial-resolution
* Pages 3-4, Sub-section 2.1: Is this sensor an analog-output sensor or a digital-output sensor with on-chip ADC? In conjunction with Figure 6 that shows the pixel values in 10^4 LSB, please add a comment on the overall sensor architecture (with or without on-chip ADC) and a comment on the magnitude of the vertical axis (i.e., x10^4 LSB) in Figure 6.
* Page 4, Table 1: About "Maximum exposure code length," I interpret the "256 bits" can be used for each tap. What is a reason why the 32-bit length was chosen for the experiments? Also, is there an optimal code length? Your additional comments on these points could be interesting for the reader.
* Page 4, Table 1: Image readout frame rate of 21 fps corresponds to 47.6ms frame time, while it seems the total exposure time is much shorter (ex. 100 x 1.65ns x 32 = 528ns). Am I wrong? Can the authors add a comment on what limits the frame rate?
* Page 9, Lines 259 - 261: This sentence ("Figure 9(a) and (b) compare ...") is not aligned with Figure 9 and the caption for it. Please correct.
* Page 4, Lines 121 - 123 and Figure 3: The term "cross-correlation" is used, but the cross-corelation function is usually defined by
y(tau) = Integral{w(t)*x(t + tau)} dt
Therefore, I am not sure if the term "cross-correlation" is appropriate. Please consider this point.
* Page 5, Line 149: The same comment on "cross-correlation," mentioned above.
Round 2
Reviewer 1 Report
Comments and Suggestions for Authors
I believe the author has adequately addressed the comments and made revisions accordingly. I have no additional comments to add. I would recommend the author to further polish the paper before considering it for publication.